# ES-dLLM: Efficient Inference for Diffusion Large Language Models by Early-Skipping

**Zijian Zhu**
IIIS, Tsinghua University
`zhuzj23@mails.tsinghua.edu.cn`

**Fei Ren**
IIIS, Tsinghua University
`renf25@mails.tsinghua.edu.cn`

**Zhanhong Tan**
Polar Bear Tech.
`tanzhh515@gmail.com`

**Kaisheng Ma** [*]
IIIS, Tsinghua University
`kaisheng@mail.tsinghua.edu.cn`

## Abstract

Diffusion large language models (dLLMs) are emerging as a promising alternative to autoregressive models (ARMs) due to their ability to capture bidirectional context and the potential for parallel generation. Despite the advantages, dLLM inference remains computationally expensive as the full input context is processed at every iteration. In this work, we analyze the generation dynamics of dLLMs and find that intermediate representations, including key, value, and hidden states, change only subtly across successive iterations. Leveraging this insight, we propose **ES-dLLM**, a training-free inference acceleration framework for dLLM that reduces computation by skipping tokens in early layers based on the estimated importance. Token importance is computed with intermediate tensor variation and confidence scores of previous iterations. Experiments on LLaDA-8B and Dream-7B demonstrate that ES-dLLM achieves throughput of up to 226.57 and 308.51 tokens per second (TPS), respectively, on an NVIDIA H200 GPU, delivering $5.6\times$ to $16.8\times$ speedup over the vanilla implementation and up to $1.85\times$ over the state-of-the-art caching method, while preserving generation quality. The source code is available at `https://github.com/zhuzj19/ES-dLLM`.

## 1 Introduction

Autoregressive models (ARMs) have been the dominant paradigm in large language models (LLMs), achieving remarkable success in a wide range of applications (Cao et al., 2023). ARMs generate text in a left-to-right pattern, producing tokens sequentially. Recently, diffusion-based LLMs (dLLMs) have emerged as a promising alternative. Unlike ARMs, dLLMs generate text through an iterative denoising process over a sequence of mask tokens, enabling bidirectional attention and offering the potential for parallel decoding (Yu et al., 2025). Industrial dLLMs such as Mercury (Inception Labs et al., 2025) and Gemini Diffusion (Deepmind, 2025) have drawn significant attention for their ultrafast generation speed. Despite this progress, open-source dLLMs, such as LLaDA (Nie et al., 2025) and Dream (Ye et al., 2025), remain far less efficient, even slower than ARMs of comparable size.

A key factor contributing to the inefficiency of dLLMs is that each iteration processes the entire sequence as input, introducing substantial computational overhead. To mitigate this cost, recent studies (Ma et al., 2025; Wu et al., 2025; Liu et al., 2025) have proposed techniques such as caching and parallel decoding mechanisms to improve generation efficiency. Nevertheless, more opportunities remain for further acceleration. In each iteration during dLLM generation, only one or a few tokens with the highest confidence are unmasked, while the majority of mask tokens are processed without producing useful results. Moreover, since the inputs of adjacent iterations differ only in the positions of newly unmasked tokens, the intermediate states of most tokens remain almost unchanged. Despite this redundancy, conventional inference procedures still compute logits for all token positions, leading to excessive and unnecessary computation.

---

[*]Corresponding Author.

To better understand these inefficiencies, we conduct a series of experiments to analyze the characteristics during the dLLM generation process. The results reveal that both intermediate tensors and confidence scores exhibit only subtle variation across successive iterations. Therefore, we identify opportunities to predict the importance of token positions and to eliminate redundant computation in early layers for tokens that contribute little to the outcome.

Motivated by these observations, we propose **ES-dLLM**[1], a training-free inference acceleration framework designed for diffusion LLMs. Specifically, it focuses on optimizing redundant token computation in each iteration, estimating token importance using intermediate variations and prior confidence scores, and skipping low-importance positions in early layers of the inference process. ES-dLLM consists of two key components:

- **Importance Score Estimation**: ES-dLLM skips irrelevant token positions by estimating their importance scores in early layers of the model, based on the variation of intermediate tensors and confidence from previous iterations. Using importance scores, only the top-$k$ positions are selected for further inference, while the remaining positions are skipped in the current iteration.

- **Partial Cache Update and Early Skip**: ES-dLLM maintains intermediate tensors as caches to be reused in subsequent iterations, including key and value tensors for attention layers and hidden states as the indicator for variation estimation in importance score. Since only a subset of tokens passes through the inference, caches corresponding to the selected positions are updated with an in-place scatter operation, while the others are reused without recomputation.

The experimental results show that ES-dLLM substantially accelerates inference, achieving throughput of up to 226.57 and 308.51 tokens per second (TPS) on the LLaDA-8B and Dream-7B models, respectively, using an NVIDIA H200 GPU, without compromising generation quality.

In summary, this paper makes the following contributions:

1. We analyze the characteristics during the dLLM generation process and observe that **intermediate tensors and confidence scores of most positions exhibit only minor variation across iterations**, thereby revealing opportunities to eliminate redundant computation.

2. We propose **ES-dLLM, a training-free inference acceleration framework that reduces per-iteration computation** by early-skipping low-importance token positions.

3. We conduct extensive experiments and ablation studies, demonstrating that **ES-dLLM achieves significant speedups of 5.6×-16.8× over the original implementation and up to 1.85× compared to state-of-the-art caching methods, all without sacrificing generation quality**.

## 2 PRELIMINARY ON DIFFUSION LARGE LANGUAGE MODELS

Diffusion large language models introduce a new paradigm in natural language processing that adopts the diffusion mechanism. Unlike traditional autoregressive models, which generate text sequentially in a token-by-token manner, dLLMs go through a series of denoising iterations, progressively unmasking tokens appended to the input.

Let $\mathcal{V}$ denote the vocabulary that includes a special mask token $[\texttt{mask}] \in \mathcal{V}$. Given an input sequence $\boldsymbol{x} = (x_1, x_2, \ldots, x_n)$ where $x_i \in \mathcal{V}$, text generation begins by appending $l$ mask tokens to form $\boldsymbol{x}^{(0)} = (x_1, x_2, \ldots, x_n, [\texttt{mask}], \ldots, [\texttt{mask}])$, where $l$ specifies the desired generation length. In the $t$-th iteration, a Transformer-based token predictor $f$ is used to estimate the probability distribution over the vocabulary for **all token positions** (rather than just the next token as in ARMs) in the sequence.

$$p_\theta(\hat{\boldsymbol{x}}^{(t)}|\boldsymbol{x}^{(t-1)}) = f(\boldsymbol{x}^{(t-1)}; \theta)$$

---

[1]The name is inspired by early-exit (EE) (Chen et al., 2024), which accelerates inference by exiting "easy" tokens early via trained EE layers. In contrast, ES-dLLM early-skip (ES) irrelevant positions without additional training.

From this distribution, a subset of mask tokens is replaced with sampled tokens, producing the updated sequence $x^{(t)}$. Several replacement strategies exist, and a widely used approach is to unmask positions with the highest confidence (Yu et al., 2025), where confidence for each position is defined as the maximum probability over the vocabulary. This iterative process continues until all mask tokens are replaced.

Compared to ARMs, which condition predictions solely on preceding tokens, dLLMs leverage the entire sequence context in each iteration. This enables them to capture bidirectional dependencies and generate more coherent text. However, it also introduces computational challenges due to bidirectional attention and flexible, non-sequential generation order. At the same time, computing entire logits creates opportunities for parallel decoding Wu et al. (2025).

## 3 RELATED WORKS FOR DLLM INFERENCE ACCELERATION

**Semi-autoregressive Generation.** LLaDA (Nie et al., 2025) adopts a semi-autoregressive generation strategy that partitions the output sequence into multiple blocks, each generated in a diffusion manner while preserving sequential order across blocks. This approach constrains the generation order, but provides better performance. Furthermore, Arriola et al. (2025) trained the BD3-LM model, which applies unidirectional attention between output blocks. This design enables lossless Key-Value (KV) caching of previous blocks, but sacrifices the ability to exploit bidirectional context.

**Accelerating using KV Caching.** In ARMs, KV caching is a common technique (Pope et al., 2023), where intermediate key and value tensors are stored and reused across decoding steps, since attention depends solely on past tokens. In diffusion LLMs, however, the situation is more challenging due to the bidirectional attention: newly unmasked tokens can influence all preceding tokens. To mitigate computational cost, several works have explored approximate KV caching strategies. dKV-Cache (Ma et al., 2025) delays the KV update for newly unmasked tokens to reduce errors, while dLLM-Cache (Liu et al., 2025) caches all intermediate tensors and adaptively updates them in each layer using a V-verify mechanism. However, these methods do not leverage the semi-autoregressive generation paradigm, which is beneficial for both efficiency and generation quality. Fast-dLLM (Wu et al., 2025) proposed DualCache for semi-autoregressive generation, which maintains separate KV caches for context on both sides and computes attention only for the current block during inference.

**Sparse Attention.** Recent works such as Sparse-dLLM (Song et al., 2025) and DPad Chen et al. (2025) exploit sparsity in attention scores and accelerate inference by modifying the set of attended tokens, either by sparsifying historical tokens or dropping distant suffix tokens. The techniques are orthogonal to our approach, which optimizes the set of processed tokens.

**Parallel Decoding.** In diffusion LLMs, logits are computed for all token positions in each iteration, allowing multiple tokens to be unmasked simultaneously. Wu et al. (2025) proposed confidence-aware parallel decoding, which dynamically adjusts the number of tokens to be unmasked per iteration based on a confidence threshold. This strategy effectively reduces the number of iterations for generation while preserving quality.

## 4 OBSERVATIONS ON DIFFUSION LLM CHARACTERISTICS

We conduct a series of experiments to analyze the characteristics of diffusion LLMs during generation, with particular attention to variations across successive iterations. For this study, we use two pre-trained diffusion LLMs, LLaDA-8B-Instruct (Nie et al., 2025) and Dream-7B-Instruct (Ye et al., 2025)[2] and perform inference on sequences sampled from widely used benchmark datasets.

### 4.1 CONFIDENCE VARIATION ACROSS ITERATIONS

In dLLM generation, a common strategy for selecting tokens to be unmasked is to choose positions with the highest confidence (i.e., the maximum softmax probability over the vocabulary for each position). Figure 1a visualizes the confidence variation, measured as the absolute difference between consecutive iterations, during generation for a sample from the BBH dataset using LLaDA-

---

[2]Results for Dream-7B-Instruct are provided in Appendix A.2.

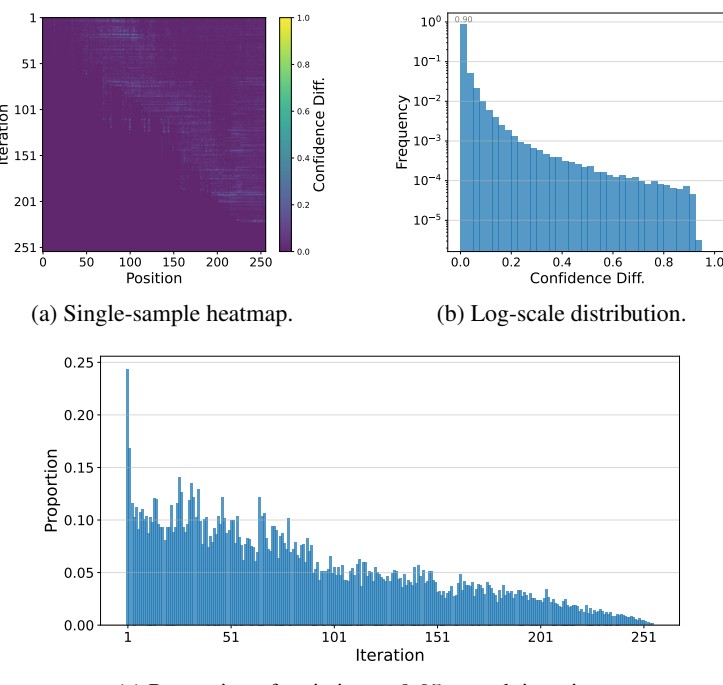

(a) Single-sample heatmap.

(b) Log-scale distribution.

(c) Proportion of variation $> 0.05$ at each iteration.

Figure 1: Confidence variation statistics using LLaDA-8B-Instruct. (a) uses a sample from the BBH dataset, while (b) and (c) present results using 100 samples from multiple datasets.

8B-Instruct. We observe that the confidence variation is subtle for most cases. To quantify this observation, Figures 1b and 1c present the distribution of confidence changes across all positions and iterations, and the proportion of positions with confidence variation greater than 0.05 at each iteration, based on 100 samples from multiple datasets. The results reveal that confidence changes approximately follow an exponential distribution, with the majority concentrated near zero. Except for some initial iterations, fewer than 10% of positions show confidence variation greater than 0.05.

The results suggest that most tokens experience only minimal confidence fluctuations across iterations. Consequently, **confidence scores from previous iterations can serve as reliable predictors, helping identify the positions that are likely to be unmasked in the current iteration**.

## 4.2 HIDDEN STATE VARIATION

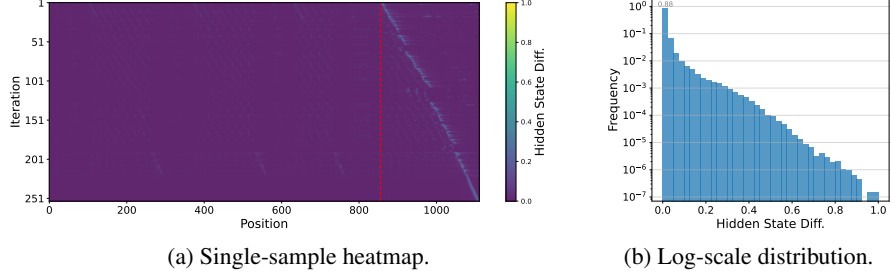

(a) Single-sample heatmap.

(b) Log-scale distribution.

Figure 2: Hidden state variation in layer 10 using LLaDA-8B-Instruct. (a) uses a single sample from the BBH dataset, while (b) presents results using 100 samples from multiple datasets. The red vertical line in (a) separates prompt and output tokens, and the distribution in (b) includes only output tokens.

Since the inputs of adjacent iterations differ only by the token that has just been unmasked, we further analyze how this input change affects the intermediate tensors. Specifically, we measure the variation of hidden states (i.e., the output after the feed-forward of a Transformer block) in a selected

Transformer layer between consecutive iterations. The variation is quantified as the normalized L1-norm of the difference, consistent with the latter term in Equation 1.

As shown in Figure 2, **the variation is minor for most positions, and only a small fraction of positions exhibit noticeable changes**. This indicates that input changes generally have a limited impact on intermediate states. Consequently, it is feasible to skip computation for tokens with small variations without degrading generation quality. Similar patterns are observed for other layers and intermediate tensors, as detailed in Appendix A.1.

## 5 METHODOLOGY

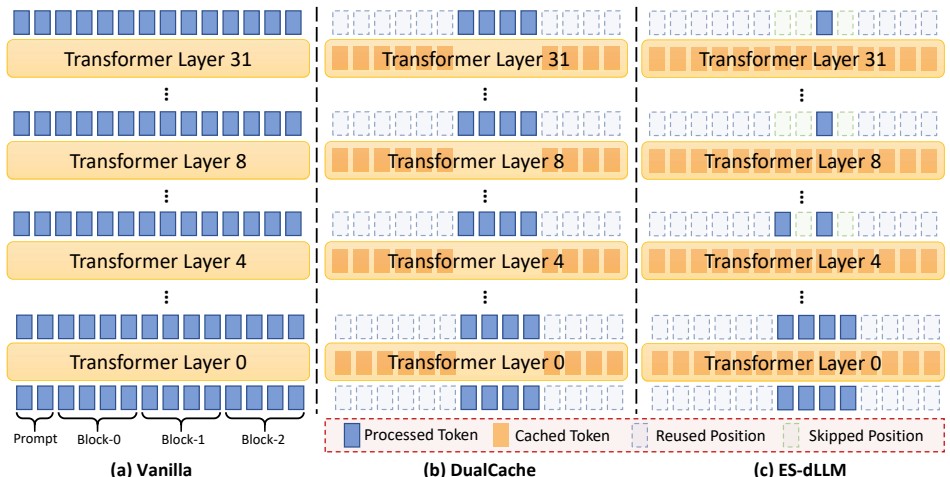

Figure 3: Illustration of ES-dLLM compared with the vanilla implementation and DualCache, assuming block-1 is under processing. The figure presents only 4 tokens per block, while the actual block length can be much larger (e.g., 32 or 64).

Motivated by the observations in Section 4, we propose **ES-dLLM**, a training-free inference acceleration framework for diffusion LLM. ES-dLLM reduces computational overhead by selectively skipping redundant token computations during inference. Specifically, it estimates the importance score for each token position using both the confidence from previous iterations and the variation of intermediate tensors, and applies an early-skip mechanism to bypass tokens with low importance in early layers. To enable this process, ES-dLLM maintains a cache that stores the necessary intermediate tensors and confidence scores for all positions. The cache is partially updated only for non-skipped tokens in each layer, while the others are reused directly. Figure 3 compares the inference process of ES-dLLM with the vanilla implementation and the state-of-the-art DualCache method. DualCache caches key and value tensors for tokens outside the current processing block, thus computing for the entire block. In contrast, ES-dLLM further reduces computation by skipping unimportant positions within the block at selected Transformer layers, achieving efficiency gains beyond DualCache.

### 5.1 IMPORTANCE SCORE ESTIMATION

We estimate the importance score of each position to determine which tokens to retain for computation. The score is calculated based on two criteria: First, mask tokens with higher confidence scores are more likely to be selected for unmasking in subsequent iterations; thus, we prefer to compute the results for these positions. Second, tokens that exhibit larger variations in intermediate tensors reflect semantic or positional dependencies on newly generated tokens; updating these positions helps capture contextual changes and mitigate error accumulation.

Formally, let the confidence score of position $i$ at iteration $t$ be denoted by $c_i^{(t)}$. The intermediate tensor, selected as the *variation indicator*, which can be query, key, value tensors, or hidden states of position $i$ in layer $l$ at iteration $t$, is denoted as $\mathbf{H}_{l,i}^{(t)}$. In layer $l$, the importance score of position $i$

at iteration $t$ is calculated as

$$I_{l,i} = \alpha \cdot c_i^{(t-1)} + (1 - \alpha) \cdot \frac{||\mathbf{H}_{l,i}^{(t)} - \mathbf{H}_{l,i}^{(t-1)}||_1}{\sqrt{d} \cdot ||\mathbf{H}_{l,i}^{(t-1)}||_2} \tag{1}$$

where $\alpha$ is a hyperparameter that weighs two criteria and $d$ is the hidden dimension of $\mathbf{H}$. The variation term is measured as the L1-norm of the difference in indicator tensors across adjacent iterations, normalized by the L2-norm in the last iteration and by $\sqrt{d}$ to align the scale. In this work, we adopt the hidden state of each Transformer layer as the indicator tensor $\mathbf{H}$. Section 6.3 also provides an ablation study that compares the choices of the variation indicator.

## 5.2 PARTIAL CACHE UPDATE AND EARLY SKIP

---

**Algorithm 1** ES-dLLM Inference Process within Transformer Block $l$

---

**Require:** Layer input $\mathbf{x}^{(l)}$ on position set $\mathbb{S}$, skip ratio $r_l$, caches of key, value, hidden states, and confidence $\mathcal{C}_\mathbf{K}, \mathcal{C}_\mathbf{V}, \mathcal{C}_\mathbf{H}, \mathcal{C}_{\mathbf{c}}$, layer index $l$
**Ensure:** Layer output $\mathbf{x}^{(l+1)}$ on updated position set $\mathbb{S}'$, updated caches
  1: $\mathbf{x}_{\text{norm}} \leftarrow \text{Norm}(\mathbf{x}^{(l)})$
  2: $\mathbf{Q}_\mathbb{S}, \mathbf{K}_\mathbb{S}, \mathbf{V}_\mathbb{S} \leftarrow \text{Q\_proj}(\mathbf{x}_{\text{norm}}), \text{K\_proj}(\mathbf{x}_{\text{norm}}), \text{V\_proj}(\mathbf{x}_{\text{norm}})$
  3: Update caches $\mathcal{C}_\mathbf{K}[l]_\mathbb{S} \leftarrow \mathbf{K}_\mathbb{S}, \mathcal{C}_\mathbf{V}[l]_\mathbb{S} \leftarrow \mathbf{V}_\mathbb{S}$ for positions in $\mathbb{S}$ using scatter operation
  4: $\mathbf{K}, \mathbf{V} \leftarrow \mathcal{C}_\mathbf{K}[l], \mathcal{C}_\mathbf{V}[l]$
  5: $\mathbf{AttnOut}_\mathbb{S} \leftarrow \text{Attention}(\mathbf{Q}_\mathbb{S}, \mathbf{K}, \mathbf{V})$
  6: $\mathbf{AttnOut}_\mathbb{S} \leftarrow \mathbf{AttnOut}_\mathbb{S} + \mathbf{x}^{(l)}$
  7: $\mathbf{H}_\mathbb{S} \leftarrow \text{FFN}(\text{Norm}(\mathbf{AttnOut}_\mathbb{S})) + \mathbf{AttnOut}_\mathbb{S}$
  8: Update cache $\mathcal{C}_\mathbf{H}[l]_\mathbb{S} \leftarrow \mathbf{H}_\mathbb{S}$
  9: Initialize importance score list $\boldsymbol{I} \leftarrow []$
10: **for** position $i$ in $\mathbb{S}$ **do**
11:     Calculate $\boldsymbol{I}[i] = I_{l,i}$ using Equation 1, with $c_i^{(t-1)} = \mathcal{C}_{\mathbf{c}i}, \mathbf{H}_{l,i}^{(t-1)} = \mathcal{C}_\mathbf{H}[l]_i, \mathbf{H}_{l,i}^{(t)} = \mathbf{H}_i$
12: **end for**
13: Select tokens of top-$k$ ($k = (1 - r_l)|\mathbb{S}|$) importance based on $\boldsymbol{I}$, denote positions as $\mathbb{S}'$
14: $\mathbf{x}^{(l+1)} \leftarrow \mathbf{H}_{\mathbb{S}'}$
15: **return** $\mathbf{x}^{(l+1)}, \mathbb{S}'$

---

We apply skipping for positions with low importance scores to reduce computational overhead; hence, only a subset of tokens is processed in each Transformer block. Therefore, we need to compute importance scores for position selection and update caches for subsequent usage. The inference process of a Transformer block with early-skipping is outlined in Algorithm 1. Given the input tensor $\mathbf{x}^{(l)}$ and the corresponding token position set $\mathbb{S}$ fed into layer $l$, we first project the normalized input to query, key, and value tensors. We use them to update the KV cache for positions in $\mathbb{S}$ and then retrieve the full KV for attention. The attention output is passed through the feed-forward layer to obtain hidden states $\mathbf{H}_\mathbb{S}$, which are also used to calculate importance scores. We select the positions with top-$(1-r_l)|\mathbb{S}|$ scores, where $r_l$ is the skip ratio that controls the fraction of skipped tokens at layer $l$. The output tensor, coupled with the updated position set, is propagated to the next layer.

The pseudo-code illustrates the case where hidden states are used as the variation indicator for importance score calculation. If other tensors, such as query, key, or value, are chosen, the importance computation and cache update steps are adjusted accordingly.

The early-skip mechanism can reduce the size of intermediate tensors by a ratio of $r_l$, which proportionally decreases the computational cost of matrix multiplication in all subsequent layers. Therefore, skipping in the earlier layer could save more FLOPs, but the reliability of tensor variation is generally better in deeper layers. We discuss this trade-off in Appendix C.2.

During generation, the cache is initialized by performing a full forward pass for all prompt and output tokens. In subsequent iterations, the model is only fed with tokens of the current block for unmasking, applying early-skipping in designated layers to skip unimportant positions and reduce overhead. To prevent error accumulation, we periodically refresh the cache for prompt tokens or the current block, where these tokens go through the entire inference process without skipping.

# 6 EXPERIMENTS

## 6.1 EXPERIMENTAL SETUP

The experiments are conducted on an NVIDIA H200 GPU. We evaluate ES-dLLM on two representative diffusion LLMs: LLaDA-8B and Dream-7B. Performance is assessed in terms of both generation quality and throughput improvement across five benchmark datasets spanning diverse LLM tasks: GSM8K (Cobbe et al., 2021) and MATH (Hendrycks et al., 2021) for mathematical questions, HumanEval (Chen et al., 2021) and MBPP (Austin et al., 2021) for code generation, and BBH (Suzgun et al., 2022) for commonsense reasoning. All evaluations are carried out using the LM-Eval framework (Gao et al., 2024).

Unless otherwise specified, the skip ratio in ES-dLLM is set to 0.5 at positions of 1/8 and 1/4 of all layers (i.e., $r_4 = r_8 = 0.5$ for LLaDA and $r_4 = r_7 = 0.5$ for Dream, while other $r_i = 0$). This configuration reduces approximately 60% of total FLOPs during inference. The parameter $\alpha$ is set to 0.5, assigning equal importance to confidence and tensor variation. To mitigate error accumulation across iterations, we refresh the caches for all preceding tokens or all tokens in the current block with a specific period, respectively, motivated by Liu et al. (2025). The settings of generation length and block length for each benchmark refer to those in LLaDA. And we use a batch size of 8 for better weight reuse and hardware utilization. More details are provided in Appendix B.

We compare ES-dLLM with two baseline methods. The first is the original implementation of LLaDA and Dream. The second is DualCache from Fast-dLLM (Wu et al., 2025), which maintains KV caches for tokens outside the current block to allow cache reuse in the attention layer and refreshes the entire cache after processing each block.

## 6.2 MAIN RESULTS

Table 1: Performance comparison using LLaDA-8B-Instruct on five benchmark datasets. TPS (tokens per second) measures throughput, calculated as the generated token count divided by total time. Speedup reports relative throughput improvement in terms of TPS compared to the vanilla implementation (i.e., LLaDA). Performance score in percentage indicates the accuracy or pass rate. The number in parentheses of each benchmark indicates the number of shots for evaluation. Bold numbers highlight the best performance on each benchmark.

| Benchmark | Method | TPS | Speedup | Performance Score |
|---|---|---|---|---|
| GSM8K(5) | LLaDA | 8.56 | 1.0× | 76.72 |
| | DualCache | 112.15 | 13.1× | 76.35 |
| | **ES-dLLM** | **143.93** | **16.8×** | **76.95** |
| MATH(4) | LLaDA | 14.04 | 1.0× | **28.14** |
| | DualCache | 56.02 | 4.0× | 26.94 |
| | **ES-dLLM** | **103.63** | **7.4×** | 27.24 |
| BBH(3) | LLaDA | 11.06 | 1.0× | **56.75** |
| | DualCache | 130.14 | 11.8× | 53.29 |
| | **ES-dLLM** | **159.89** | **14.5×** | 54.51 |
| | ES-dLLM* | 133.48 | 12.1× | 56.66 |
| HumanEval(0) | LLaDA | 23.65 | 1.0× | 36.59 |
| | DualCache | 176.85 | 7.5× | 35.37 |
| | **ES-dLLM** | **226.57** | **9.6×** | **37.8** |
| MBPP(3) | LLaDA | 8.98 | 1.0× | **41** |
| | DualCache | 117.85 | 13.1× | 38.4 |
| | **ES-dLLM** | **145.99** | **16.3×** | 39.4 |

Tables 1 and 2 present the performance comparison of ES-dLLM with the baselines on LLaDA-8B-Instruct and Dream-7B-Instruct, respectively. ES-dLLM achieves substantial efficiency gains, with speedups ranging from 5.6× to 16.8× over the original implementations and from 1.20× to

Table 2: Performance comparison using Dream-7B-Instruct on five benchmark datasets. Metrics and notations are the same as in Table 1.

| Benchmark | Method | TPS | Speedup | Performance Score |
|---|---|---|---|---|
| GSM8K(5) | Dream | 19.80 | 1.0× | **79.00** |
| | DualCache | 209.88 | 10.6× | 77.94 |
| | **ES-dLLM** | **267.13** | **13.5×** | 77.94 |
| MATH(4) | Dream | 26.38 | 1.0× | **33.94** |
| | DualCache | 86.38 | 3.3× | 33.60 |
| | **ES-dLLM** | **147.44** | **5.6×** | 33.44 |
| BBH(3) | Dream | 24.84 | 1.0× | **61.48** |
| | DualCache | 226.89 | 9.1× | 58.27 |
| | **ES-dLLM** | **292.23** | **11.8×** | 57.84 |
| | **ES-dLLM*** | 196.50 | 7.9× | 60.36 |
| HumanEval(0) | Dream | 44.34 | 1.0× | **46.95** |
| | DualCache | 258.10 | 5.8× | 45.12 |
| | **ES-dLLM** | **308.51** | **7.0×** | 45.12 |
| MBPP(3) | Dream | 21.68 | 1.0× | **60.4** |
| | DualCache | 214.14 | 9.9× | 56.8 |
| | **ES-dLLM** | **276.12** | **12.7×** | 57 |
| | **ES-dLLM*** | 130.49 | 6.0× | 59 |

$1.85\times$ over DualCache. In terms of generation quality, ES-dLLM generally matches DualCache and even surpasses it on several benchmarks. This result suggests that frequent updates of a larger set of tokens may not always be beneficial and may introduce noise. These findings highlight both the computational redundancy inherent during dLLM generation and the effectiveness of ES-dLLM's early-skip strategy.

Moreover, we observe that DualCache suffers from a degraded accuracy on the BBH and MBPP datasets compared to the original implementation, which we attribute to error accumulation in the KV cache of the prompt region. To address this, we evaluate ES-dLLM with more frequent KV cache refreshment for prompt tokens (i.e., multiple times within each block), denoted as **ES-dLLM***. As reported in the tables, this adjustment effectively mitigates the accuracy drop while retaining decent speedups. Overall, **ES-dLLM delivers a speedup of at least 5.6×, with accuracy differences ranging between -1.83 and +1.21 compared to the vanilla implementations**, demonstrating both the efficiency and robustness of ES-dLLM in accelerating diffusion LLM inference.

## 6.3 ABLATION STUDY

**Effect of $\alpha$ in Importance Score Estimation.** The parameter $\alpha$ in Equation 1 is used to weigh the contributions of confidence and variation in importance scores. We evaluate different $\alpha$ values on LLaDA-8B-Instruct across multiple benchmarks, with the results shown in Figure 4a. It indicates that combining both factors ($\alpha = 0.5$) achieves the best overall performance, while relying on one of them leads to noticeable accuracy degradation, particularly when using the confidence score alone ($\alpha = 1$), which likely stems from neglecting contextual updates from previously generated tokens. Interestingly, $\alpha = 0$ (i.e., using only variation scores) performs well on the MATH dataset, even surpassing the original baseline, suggesting that the generation process may not strictly follow the confidence order for this task.

**Selection of Intermediate Tensors for Variation Indicator.** In our main experiments, we use hidden states as the variation indicator. To evaluate the sensitivity of this choice, we compare alternative indicators, including key, query, and value tensors, in Figure 4b. The results show that generation quality is generally robust to the choice of variation indicator. Hidden states provide slightly better performance overall, but key and value tensors achieve comparable results with lower memory overhead, offering a practical alternative.

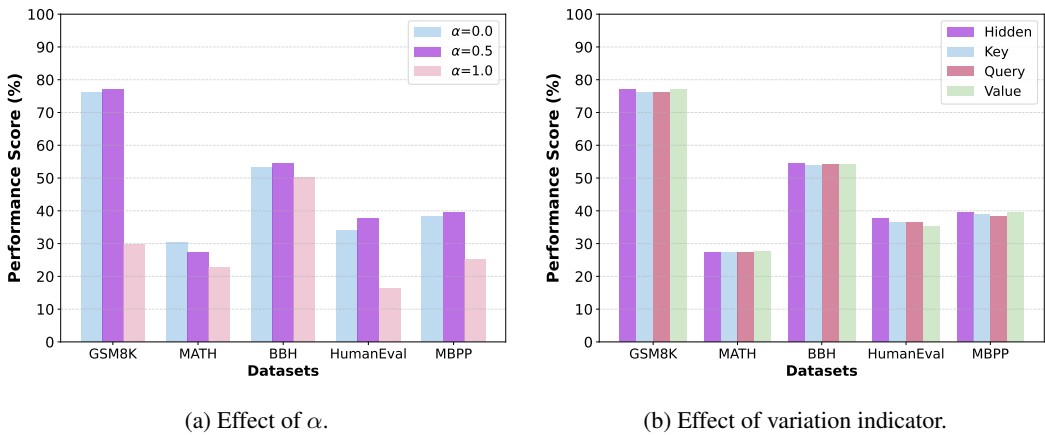

(a) Effect of $\alpha$.  (b) Effect of variation indicator.

Figure 4: Ablation studies on importance estimation configurations using LLaDA-8B-Instruct.

Additional analyses, including results on Base models, ablations on skip ratio and position, and the compatibility of ES-dLLM with sparse attention and parallel decoding techniques, are presented in Appendix C.

## 7 DISCUSSION

**Memory Overhead of ES-dLLM.** ES-dLLM maintains caches for key, value, and variation indicator (e.g., hidden state) tensors for each token. The cache of the variation indicator is only needed for output positions in layers where skipping is applied. Therefore, the additional memory overhead of ES-dLLM is 528KB for LLaDA-8B and 70KB for Dream-7B per output token in BF16 format, of which hidden states contribute only 16KB and 14KB, respectively. Considering a sample with 1024 prompt tokens and a generation length of 256, the total memory overhead of ES-dLLM is 644MB for LLaDA-8B and 73.5MB for Dream-7B per sample. This cost is modest and with negligible overhead beyond the KV caches, which is acceptable for modern GPUs with large memory capacities, especially in comparison to the over 10GB required for model weights.

**Speedup Potential of ES-dLLM.** With the chosen skip configuration, ES-dLLM reduces FLOPs by approximately 60% compared to DualCache. However, the observed speedup ranges from 1.20× to 1.85×, smaller than the theoretical computation reduction. This discrepancy can be attributed to the fact that LLM inference with fewer tokens (e.g., the decode phase in autoregressive LLM) is often memory-bound rather than compute-bound. According to the roofline model (Williams et al., 2009), efficiency is constrained by memory bandwidth and operational intensity. While ES-dLLM lowers FLOPs by skipping computation, memory accesses for model weights and KV tensors remain largely unchanged, shifting the workload from compute-bound to memory-bound. Therefore, this provides opportunities for system-level optimizations (Patel et al., 2024; Agrawal et al., 2024) that better integrate compute-bound and memory-bound workloads, potentially unlocking the full speedup potential of ES-dLLM.

**Limitation and Future Work.** Although ES-dLLM effectively exploits redundancy in token computation, the estimation of the importance score relies on simple heuristics, and partial KV updating diverges from the training process for dLLM that assumes complete state updates. Future work could explore more sophisticated importance estimation and skipping methods, such as training a lightweight model to predict token importance, adaptively adjusting skip ratios based on real-time variation, and investigating training strategies that align with the early-skipping mechanism to further enhance both efficiency and generation quality.

## 8 CONCLUSION

In this work, we propose ES-dLLM, a training-free inference acceleration framework for diffusion large language models that reduces computational overhead by selectively skipping redundant com-

putations in early layers. ES-dLLM achieves up to $16.8\times$ speedup over the original implementation and up to $1.85\times$ speedup compared to the state-of-the-art DualCache method on LLaDA-8B and Dream-7B models, while maintaining comparable generation quality. Our results reveal redundancy in token computation during dLLM generation, and ES-dLLM provides a practical direction to optimize the inference process of dLLM.

## REPRODUCIBILITY STATEMENT

For reproducibility, we open-source our code in `https://github.com/zhuzj19/ES-dLLM`, accompanied by a README file with detailed instructions. The experimental settings are described in Section 6.1 and Appendix B.1. In addition, we provide bash scripts in the supplementary material for reproducing the main results reported in this paper.

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

## A ADDITIONAL EXPERIMENTS ON DLLM GENERATION CHARACTERISTICS

### A.1 INTERMEDIATE TENSOR VARIATION IN LLADA

In addition to hidden states, we also examine the variation of other intermediate tensors, namely the key, value, and query tensors inside the attention. As shown in Figure 5, their variations are relatively small for most tokens in most iterations, consistent with the observations on hidden states discussed in Section 4.2.

We further plot the variation distribution of hidden states in other layers in Figure 6, which shows exponential distributions as those observed for layer 10 in the main text. We perform normalization for values greater than 1, producing a slight spike at 1 in the distribution for layer 30. The variation tends to increase in deeper layers, likely because of multiple interactions among tokens in attention, which can amplify variation in intermediate results. Nevertheless, even in the layer approaching the end, a large portion of positions still exhibit small variations.

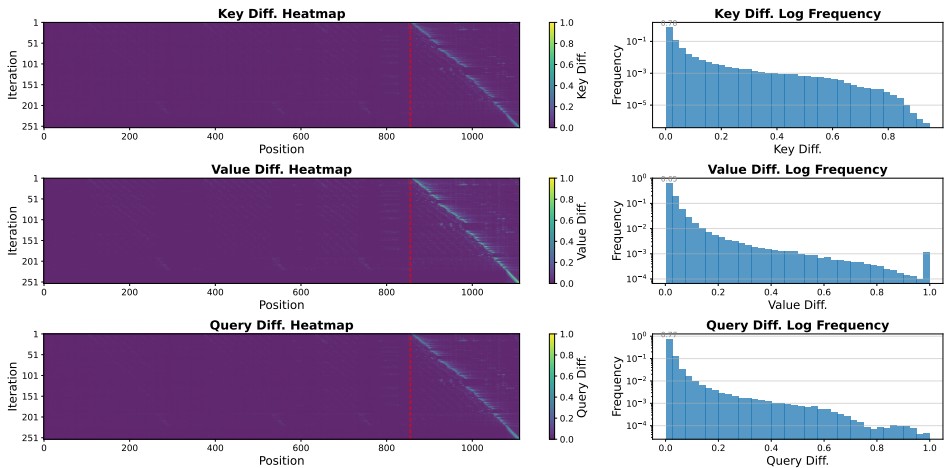

Figure 5: Variation statistics of key, value, and query tensors in layer 10 using LLaDA-8B-Instruct. Left: single-sample heatmap from BBH; red line separates prompt and output tokens. Right: log-scale distribution for output tokens using 100 samples from multiple datasets.

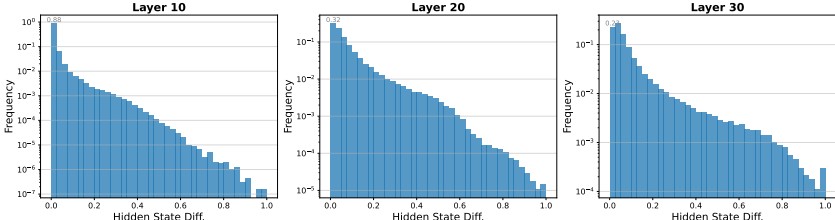

Figure 6: Log-scale distributions of hidden state variation in layers 10, 20, and 30 using LLaDA-8B-Instruct.

### A.2 EXPERIMENTS ON GENERATION CHARACTERISTICS IN DREAM

We perform the same set of experiments on Dream-7B-Instruct to validate the generality of the observations presented in Section 4. The results are shown in Figures 7 and 8. The confidence change and intermediate tensor variation are slightly larger, but follow trends similar to those observed in LLaDA-8B-Instruct, demonstrating that the characteristics are consistent across different diffusion LLMs. For the specific sample presented in the heatmaps, we observe a pronounced confidence variation at the iteration corresponding to the generation of the EOS token, which also influences the intermediate tensor variation in subsequent iterations.

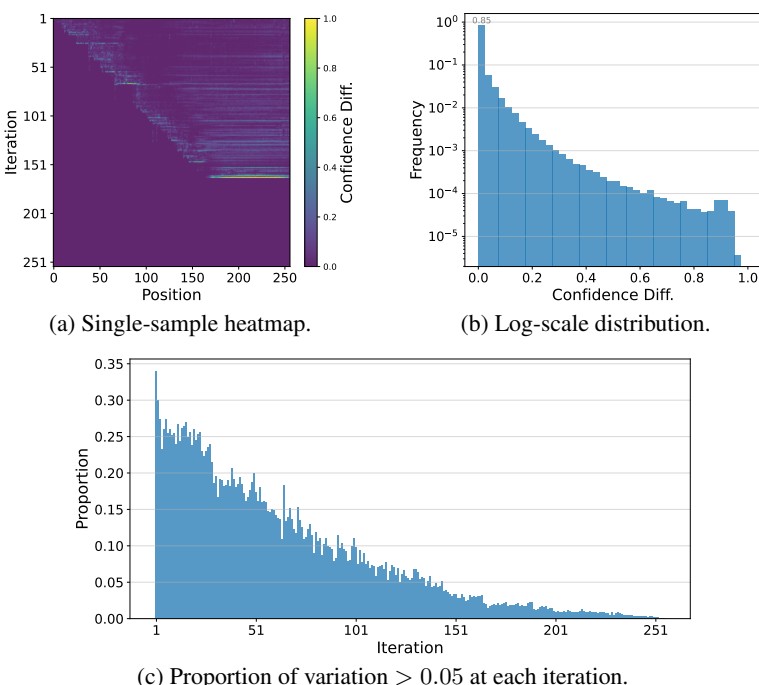

(a) Single-sample heatmap.

(b) Log-scale distribution.

(c) Proportion of variation $> 0.05$ at each iteration.

Figure 7: Confidence variation statistics using Dream-7B-Instruct. (a) uses a sample from the BBH dataset, while (b) and (c) present results using 100 samples from multiple datasets.

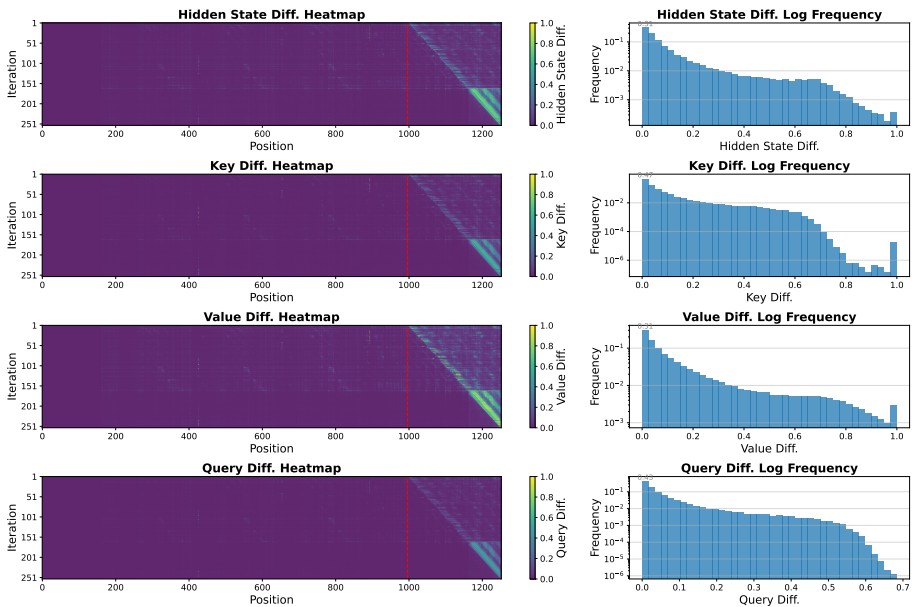

Figure 8: Variation statistics of hidden states and key, query, and value tensors in layer 10 using Dream-7B-Instruct. Left: single-sample heatmap from BBH; red line separates prompt and output tokens. Right: log-scale distribution for output tokens using 100 samples from multiple datasets.

### A.3 CORRELATION BETWEEN VARIATIONS OF INTERMEDIATE TENSOR AND CONFIDENCE

We further analyze the relationship between intermediate tensor variation and confidence difference using 100 samples from multiple datasets. As shown in Table 3, the correlation coefficients range from 0.15 to 0.66, reflecting a positive correlation. The correlation tends to be stronger in later layers, which is expected since the final predictions rely on the hidden states of the last layer. How-

Table 3: Pearson correlation between intermediate tensor variation and probability change (confidence is the maximum probability) for all mask tokens in selected layers using LLaDA-8B-Instruct.

| Layer | 0 | 4 | 8 | 16 | 24 | 31 |
|---|---|---|---|---|---|---|
| Hidden | 0.15 | 0.22 | 0.27 | 0.38 | 0.48 | 0.51 |
| Query | N/A | 0.18 | 0.24 | 0.39 | 0.45 | 0.66 |
| Key | N/A | 0.18 | 0.24 | 0.38 | 0.42 | 0.66 |
| Value | N/A | 0.16 | 0.25 | 0.43 | 0.45 | 0.65 |

ever, applying skipping in earlier layers could save more computation, creating a trade-off between efficiency and the reliability of tensor variation as an indicator. We further discuss this trade-off by evaluating different skipping positions in Appendix C.2. Notably, the correlation is not applicable for query, key, and value tensors in layer 0, as they are directly projected from the input embeddings without inter-token interaction, thus only differ in the newly generated tokens.

## B    EXPERIMENT DETAILS

### B.1    PARAMETER SETTINGS

Table 4: Generation length and block length for each benchmark.

| Benchmark | Generation Length | Block Length |
|---|---|---|
| GSM8K | 256 | 64 |
| MATH | 256 | 256 |
| BBH | 256 | 64 |
| HumanEval | 512 | 64 |
| MBPP | 512 | 64 |

In addition to the parameters described in the main text, we provide more details of the experimental settings to ensure reproducibility. Following Nie et al. (2025), we adopt their choices of generation and block lengths for each benchmark, with some adjustments. The specific configurations are listed in Table 4. Although Dream does not inherently employ a semi-autoregressive generation strategy, we found that constraining the generation order with a block length can improve performance. Therefore, we adopt the same semi-autoregressive block setting for both models.

Table 5: Cache refresh periods in ES-dLLM for each benchmark and model.

(a) LLaDA-8B-Instruct

| Benchmark | Prompt | Block |
|---|---|---|
| GSM8K | 64 | 16 |
| MATH | 256 | 8 |
| BBH | 64 | 4 |
| HumanEval | 64 | 4 |
| MBPP | 64 | 4 |

(b) Dream-7B-Instruct

| Benchmark | Prompt | Block |
|---|---|---|
| GSM8K | 64 | 8 |
| MATH | 256 | 4 |
| BBH | 64 | 8 |
| HumanEval | 64 | 2 |
| MBPP | 256 | 2 |

Table 6: Cache refresh periods in ES-dLLM* for each benchmark and model.

| Model | Benchmark | Prompt | Block |
|---|---|---|---|
| LLaDA | BBH | 32 | 4 |
| Dream | BBH | 16 | 4 |
| | MBPP | 8 | 2 |

Moreover, as mentioned in the main text, we periodically refresh the cache for the prompt tokens or for the entire current block during generation. We employ different cache refresh frequency settings for each benchmark and model to achieve better performance, as listed in Table 5. On the BBH and MBPP datasets, we observe a performance gap between DualCache and the original implementation. This difference arises because DualCache intrinsically applies a prompt refresh period equal to the block size and a block refresh period of 1, whereas the original implementation updates both at every step (i.e., period of 1). To mitigate this accuracy loss, we increase the prompt refresh frequency in ES-dLLM, resulting in a variant denoted as ES-dLLM* in the main text. The specific refresh period configurations of ES-dLLM* are listed in Table 6.

For the sampling strategy, we adopt low-confidence remasking for LLaDA and maskgit-plus with top-k ($k = 50$) and top-p ($p = 0.95$) sampling for Dream. The temperature is set to 0 for both models. For evaluation metrics to get the performance score of each dataset, we use `exact_match` for GSM8K and BBH (with GSM8K using the `flexible-extract` filter), `math_verify` metric for MATH, and `pass_at_1` for the two coding datasets HumanEval and MBPP.

### B.2 IMPLEMENTATION DETAILS

We implement ES-dLLM and the DualCache baseline on top of the open-source codebases of LLaDA (Nie et al., 2025) and Dream (Ye et al., 2025). To enable the early-skipping mechanism, we modify several modules, including tensor caching, importance score computation, and position selection logic. For LLaDA, we extend the framework to support batch inference. KV caching is applied after the RoPE operation to reduce the computation overhead. We notice that operating on full logits consumes a significant portion of memory during generation, especially for top-k sampling. To alleviate this issue, we truncate the logits of prompt tokens before sampling. In addition, we observe that the EOS token is sometimes generated prematurely, producing incomplete outputs across all baselines. To address this, we disallow EOS generation when the last token is still a mask token to improve stability, which we found beneficial for the performance, especially on coding datasets.

## C SUPPLEMENTARY EXPERIMENTS

### C.1 MAIN RESULTS ON BASE MODELS

Table 7: Performance comparison using LLaDA-8B-Base on five benchmark datasets. Metrics and notations follow Table 1.

| Benchmark | Method | TPS | Speedup | Performance Score |
|---|---|---|---|---|
| GSM8K(5) | LLaDA | 9.20 | 1.0× | **71.11** |
| | DualCache | 116.46 | 12.7× | 66.19 |
| | **ES-dLLM** | **140.45** | **15.3×** | 67.63 |
| MATH(4) | LLaDA | 15.12 | 1.0× | **32.38** |
| | DualCache | 57.89 | 3.8× | 30.48 |
| | **ES-dLLM** | **94.46** | **6.2×** | 30.60 |
| BBH(3) | LLaDA | 11.68 | 1.0× | **45.26** |
| | DualCache | 133.95 | 11.5× | 43.59 |
| | **ES-dLLM** | **162.30** | **13.9×** | 44.08 |
| HumanEval(0) | LLaDA | 23.84 | 1.0× | **32.32** |
| | DualCache | 177.40 | 7.4× | **32.32** |
| | **ES-dLLM** | **232.66** | **9.8×** | 31.71 |
| MBPP(3) | LLaDA | 9.53 | 1.0× | **39.6** |
| | DualCache | 120.12 | 12.6× | 38.6 |
| | **ES-dLLM** | **158.81** | **16.7×** | 38 |

We further evaluate ES-dLLM using the base models LLaDA-8B-Base and Dream-7B-Base across five benchmark datasets, with results reported in Tables 7 and 8. ES-dLLM achieves speedups

Table 8: Performance comparison of Dream-7B-Base on five benchmark datasets.

| Benchmark | Method | TPS | Speedup | Performance Score |
|---|---|---|---|---|
| GSM8K(5) | Dream | 21.12 | 1.0× | **74.15** |
| | DualCache | 211.67 | 10.2× | 73.16 |
| | **ES-dLLM** | **282.67** | **13.4×** | 73.92 |
| MATH(4) | Dream | 27.98 | 1.0× | **40.90** |
| | DualCache | 86.58 | 3.1× | 40.10 |
| | **ES-dLLM** | **167.03** | **6.0×** | 39.60 |
| BBH(3) | Dream | 25.94 | 1.0× | **49.65** |
| | DualCache | 228.09 | 8.8× | 45.34 |
| | **ES-dLLM** | **281.65** | **10.9×** | 44.28 |
| HumanEval(0) | Dream | 44.26 | 1.0× | **42.07** |
| | DualCache | 257.42 | 5.8× | 40.24 |
| | **ES-dLLM** | **305.37** | **6.9×** | 38.41 |
| MBPP(3) | Dream | 22.73 | 1.0× | **55** |
| | DualCache | 216.09 | 9.5× | 56.2 |
| | **ES-dLLM** | **301.43** | **13.3×** | 55.2 |

of up to 16.7× and 1.93× over the original implementations and DualCache, respectively, while maintaining comparable performance scores, demonstrating the effectiveness and generality of ES-dLLM on base models.

## C.2   ABLATION STUDY ON SKIP RATIO AND POSITION

Table 9: Ablation study of skip ratio and position configurations on MATH using LLaDA-8B-Instruct. No skipping represents the DualCache baseline. Speedup is calculated against the TPS of DualCache, and the FLOPs proportion is normalized to the no-skipping baseline.

| Skip Ratio & Position | FLOPs Prop. | TPS | Speedup | Performance Score |
|---|---|---|---|---|
| No skipping | 100% | 56.02 | 1.0× | 26.94 |
| $r_4 = r_8 = 0.5$ | 40% | 103.63 | 1.85× | 27.24 |
| $r_8 = 0.75$ | **46%** | **95.26** | **1.70×** | 27.26 |
| $r_8 = 0.5$ | 64% | 77.66 | 1.39× | 27.36 |
| $r_8 = 0.25$ | 82% | 63.49 | 1.13× | **27.58** |
| $r_0 = 0.5$ | **52%** | **89.40** | **1.60×** | 26.76 |
| $r_4 = 0.5$ | 58% | 83.17 | 1.48× | 27.28 |
| $r_8 = 0.5$ | 64% | 77.66 | 1.39× | 27.36 |
| $r_{16} = 0.5$ | 77% | 68.50 | 1.22× | **27.52** |

Table 10: Ablation study on skipping times across five datasets using LLaDA-8B-Instruct.

| Skip Ratio& Position | FLOPs Prop. | GSM8K | MATH | BBH | HumanEval | MBPP |
|---|---|---|---|---|---|---|
| $r_4 = 0.7$ | 40% | 76.27 | 27.44 | 53.95 | 35.98 | 39.4 |
| $r_4 = r_8 = 0.5$ | 40% | **76.95** | 27.24 | **54.51** | **37.8** | 39.4 |
| $r_4 = r_8 = r_{12} = 0.405$ | 40% | 76.35 | **27.48** | 53.86 | 36.59 | **39.8** |

In the main experiments, we apply early-skipping at two positions (1/8 and 1/4 of all layers) with a skip ratio of 0.5. To further investigate the impact of the skip ratio and skip position, we evaluate a range of configurations, with results reported in Tables 9 and 10.

As shown in Table 9, varying the skip ratio at a fixed position reveals a trade-off between efficiency and generation quality. Larger ratios yield greater speedup by saving more computation, but also adversely affect performance. A similar trade-off is observed for skip position: applying skipping in earlier layers brings greater speedup, but leads to performance degradation due to insufficient intermediate information.

We also examine the effect of applying different times of skipping while keeping the overall FLOPs proportion roughly the same (about 40%). As shown in Table 10, skipping at two positions ($r_4 = r_8 = 0.5$) achieves the best balance, progressively refining token selection and avoiding overly aggressive pruning in early layers. In contrast, applying too many skips could reduce the number of remaining tokens in the final layer, which is also detrimental to performance.

## C.3 INTEGRATION WITH EXISTING METHODS

Table 11: Performance comparison for parallel decoding using LLaDA-8B-Instruct. Speedup is compared with the DualCache baseline without parallel decoding.

| Benchmark | Method | TPS | Speedup | Performance Score |
|---|---|---|---|---|
| GSM8K(5) | DualCache+PD | 172.02 | 1.53× | **76.57** |
| | **ES-dLLM**+PD | **201.60** | **1.80×** | 75.74 |
| MATH(4) | DualCache+PD | 85.05 | 1.52× | 26.94 |
| | **ES-dLLM**+PD | **152.19** | **2.72×** | **27.48** |
| BBH(3) | DualCache+PD | 269.48 | 2.07× | 52.31 |
| | **ES-dLLM**+PD | **302.95** | **2.33×** | **53.26** |
| HumanEval(0) | DualCache+PD | 271.64 | 1.54× | 34.76 |
| | **ES-dLLM**+PD | **349.26** | **1.97×** | **37.8** |
| MBPP(3) | DualCache+PD | 367.82 | 3.12× | **38.4** |
| | **ES-dLLM**+PD | **413.82** | **3.51×** | 37.8 |

Table 12: Performance comparison for parallel decoding using Dream-7B-Instruct.

| Benchmark | Method | TPS | Speedup | Performance Score |
|---|---|---|---|---|
| GSM8K(5) | DualCache+PD | 364.48 | 1.74× | 77.94 |
| | **ES-dLLM**+PD | **425.27** | **2.03×** | **78.01** |
| MATH(4) | DualCache+PD | 128.02 | 1.48× | **33.58** |
| | **ES-dLLM**+PD | **215.61** | **2.50×** | 33.48 |
| BBH(3) | DualCache+PD | 546.51 | 2.41× | **57.18** |
| | **ES-dLLM**+PD | **615.51** | **2.71×** | 57.07 |
| HumanEval(0) | DualCache+PD | 513.85 | 1.99× | 44.51 |
| | **ES-dLLM**+PD | **710.33** | **2.75×** | **46.34** |
| MBPP(3) | DualCache+PD | 829.71 | 3.87× | 56.8 |
| | **ES-dLLM**+PD | **1469.15** | **6.86×** | **57.4** |

As mentioned in Section 3, ES-dLLM is orthogonal and complementary to existing acceleration techniques, including parallel decoding (Wu et al., 2025) and sparse attention (Song et al., 2025). In this section, we explore the compatibility and try to integrate ES-dLLM with these methods.

### C.3.1 INTEGRATION WITH PARALLEL DECODING

In the main text, we evaluate ES-dLLM under the one-token-per-step generation scheme. However, dLLM has the potential to generate multiple tokens in parallel at each iteration, offering further acceleration. To explore this, we integrate ES-dLLM with confidence-aware parallel decoding

(PD) (Wu et al., 2025), using a confidence threshold of 0.9. The cache refresh period settings for ES-dLLM follow the same configuration as in Table 5.

Tables 11 and 12 present the results. We find that ES-dLLM can be seamlessly integrated with parallel decoding, delivering additional speedups of up to $1.79\times$ for LLaDA and $1.68\times$ for Dream compared to DualCache with parallel decoding, while maintaining comparable performance scores. These results confirm that **ES-dLLM is fully compatible with parallel decoding** and can further enhance the efficiency of diffusion LLM inference.

### C.3.2 INTEGRATION WITH SPARSE ATTENTION

Table 13: Performance comparison for sparse attention using LLaDA-8B-Instruct. Speedup is compared to DualCache without sparse attention technique.

| Benchmark | Method | TPS | Speedup | Performance Score |
|---|---|---|---|---|
| GSM8K(5) | Sparse-dLLM | 140.26 | 1.25× | **76.80** |
| | **ES-dLLM**+Sparse | **172.55** | **1.54×** | 75.97 |
| MATH(4) | Sparse-dLLM | 66.80 | 1.19× | **28.10** |
| | **ES-dLLM**+Sparse | **121.91** | **2.18×** | 28.06 |
| BBH(3) | Sparse-dLLM | 160.35 | 1.23× | 49.13 |
| | **ES-dLLM**+Sparse | **191.56** | **1.47×** | **51.13** |
| HumanEval(0) | Sparse-dLLM | 207.95 | 1.18× | **35.37** |
| | **ES-dLLM**+Sparse | **261.63** | **1.48×** | 34.76 |
| MBPP(3) | Sparse-dLLM | 145.77 | 1.24× | **40.2** |
| | **ES-dLLM**+Sparse | **179.57** | **1.52×** | 38.8 |

Table 14: Performance comparison for sparse attention using Dream-7B-Instruct.

| Benchmark | Method | TPS | Speedup | Performance Score |
|---|---|---|---|---|
| GSM8K(5) | Sparse-dLLM | 227.68 | 1.08× | **77.94** |
| | **ES-dLLM**+Sparse | **287.68** | **1.37×** | **77.94** |
| MATH(4) | Sparse-dLLM | 89.56 | 1.04× | 35.60 |
| | **ES-dLLM**+Sparse | **157.19** | **1.82×** | **36.10** |
| BBH(3) | Sparse-dLLM | 243.35 | 1.07× | **57.52** |
| | **ES-dLLM**+Sparse | **305.35** | **1.35×** | 57.09 |
| HumanEval(0) | Sparse-dLLM | 270.53 | 1.05× | 45.73 |
| | **ES-dLLM**+Sparse | **322.05** | **1.25×** | **46.95** |
| MBPP(3) | Sparse-dLLM | 233.86 | 1.09× | 56.6 |
| | **ES-dLLM**+Sparse | **309.74** | **1.45×** | **57.2** |

Sparse-dLLM (Song et al., 2025) is another technique that accelerates dLLM inference by pruning KV caching of less important tokens from being attended in the attention operation, while ES-dLLM focuses on reducing the number of tokens going through inference using early skipping. To investigate their compatibility, we integrate ES-dLLM with sparse attention, using the settings from the original paper: a retention ratio of 0.5, a kernel size of 3, and a delayed step of 1.

The results in Tables 13 and 14 show that ES-dLLM can be **effectively combined with sparse attention**, achieving additional speedup while maintaining comparable performance, further validating the flexibility of ES-dLLM. Sparse-dLLM achieves less significant speedup on Dream-7B-Instruct compared to LLaDA-8B-Instruct, likely due to Dream's GQA mechanism that already reduces KV cache overhead in attention, and the proportion of attention in the overall inference time is smaller.

Table 15: Performance of ES-dLLM when combined with both parallel decoding and sparse attention.

| Benchmark | GSM8K | MATH | BBH | HumanEval | MBPP |
|---|---|---|---|---|---|
| *LLaDA-8B-Instruct* | | | | | |
| TPS | 241.03 | 175.53 | 331.30 | 407.86 | 483.35 |
| (vs. DualCache) | 2.15× | 3.13× | 2.55× | 2.31× | 4.10× |
| Performance | 74.30 | 28.46 | 52.17 | 34.76 | 38.6 |
| (vs. DualCache) | -2.05 | +1.52 | -1.12 | -0.61 | +0.2 |
| *Dream-7B-Instruct* | | | | | |
| TPS | 459.63 | 228.00 | 645.27 | 763.21 | 1618.92 |
| (vs. DualCache) | 2.19× | 2.64× | 2.84× | 2.96× | 7.56× |
| Performance | 77.63 | 35.64 | 56.47 | 45.12 | 56.8 |
| (vs. DualCache) | -0.31 | +2.04 | -1.8 | 0 | 0 |

### C.3.3 INTEGRATION WITH BOTH METHODS

Moreover, ES-dLLM can be integrated with both parallel decoding and sparse attention simultaneously to achieve full speedup potential. As shown in Tables 15, this combined approach achieves speedups of 2.15-4.10× on LLaDA-8B-Instruct and 2.19-7.56× on Dream-7B-Instruct relative to the DualCache baseline, while maintaining competitive performance scores. These findings demonstrate that redundancy in vanilla dLLM inference can be effectively exploited from multiple dimensions, and **ES-dLLM serves as a versatile component that can be seamlessly combined with other acceleration techniques to enable simple and efficient deployment**.

## D  THE USE OF LARGE LANGUAGE MODELS

In preparing this paper, we employed LLMs to assist with paper writing. Specifically, GitHub Copilot was used to help with some word and sentence drafting, and ChatGPT was used to refine and polish the language.

