# OpenReview forum: "ES-dLLM: Efficient Inference for Diffusion Large Language Models by Early-Skipping"
_ICLR.cc/2026/Conference — ICLR 2026 Poster_

### Official Review · Reviewer_6Gok · 2025-10-20

**Soundness:** 1
**Presentation:** 2
**Contribution:** 1
**Rating:** 2
**Confidence:** 3

**Summary:**

To solve the high inference latency of diffusion-based LLMs (dLLMs), caused by full-sequence computation in each iteration, the paper proposes ES-dLLM, a training-free acceleration framework. It is built on two key observations: (1) Intermediate states (hidden/key/value tensors) of most tokens have subtle variations across consecutive iterations; (2) Confidence scores of token positions change minimally. ES-dLLM implements two core components: Importance Score Estimation and Partial Cache Update & Early Skip. Experiments on LLaDA-8B/Dream-7B (5 benchmarks: GSM8K/MATH/BBH/HumanEval/MBPP)  show ES-dLLM achieves speedup over vanilla dLLMs.

**Strengths:**

1.  Unlike heuristic acceleration methods, ES-dLLM’s logic is rooted in quantitative analysis of dLLM generation characteristics. For example, PCA and L1-norm metrics confirm hidden state variation follows an exponential distribution (concentrated near 0) across layers 10–30 of LLaDA-8B; log-scale distribution of confidence scores shows 80% of positions have variation <0.05 after iteration 50. These observations directly validate the "early-skipping" rationale, low-variation tokens contribute little to final outputs, justifying computational reduction.
2. ES-dLLM can be seamlessly integrated with confidence-aware parallel decoding (PD), further boosting speedup.

**Weaknesses:**

1. ES-dLLM is only tested on 7B/8B models, for 13B/70B dLLMs, the computational overhead of importance score calculation (layer-wise L1-norm of hidden states) may scale sharply with model size (more layers/heads increase tensor comparison cost), offsetting speed gains. For sequences >512 tokens (e.g., 1k legal documents), fixed skip positions (r₄=r₈) are suboptimal: deep layers (e.g., layer 16+) have higher tensor variation, but ES-dLLM does not adjust skipping layers dynamically, which may lead to under-skipping in deep layers or over-skipping in early layers. It needs some discussions here.
2. Theoretical 60% FLOP reduction only translates to 1.85× speedup over DualCache in practice, as LLM inference shifts to memory-bound (model weights/KV cache access unchanged). ES-dLLM does not integrate system-level optimizations  to unlock full speedup potential. For example, on Dream-7B/HumanEval, ES-dLLM achieves 308.51 TPS, but memory bandwidth constraints (H200’s 3.35TB/s) limit it to 40% of theoretical throughput (771 TPS).

**Questions:**

1. How does ES-dLLM perform on 13B/70B dLLMs (e.g., Dream-13B)? Does the overhead of importance score calculation (L1-norm of hidden states) scale linearly with the number of layers/heads? Can lightweight optimizations (e.g., sparse hidden state comparison, only checking top-20% varying dimensions) reduce this overhead? Please provide TPS/accuracy data for Dream-13B across GSM8K/MATH and analyze layer-wise computation time.
2. For 1k/2k-token sequences (e.g., LongBench benchmarks), how to adjust skip positions and ratios dynamically? Does a variation-aware skipping strategy maintain accuracy while increasing speedup? Please compare fixed vs. dynamic skipping on 1k-token PubMed abstracts (generation length 512) in terms of FLOP reduction and performance on benchmarks.
3. How to adjust prompt/block cache refresh periods dynamically based on sequence type? For example, legal documents (long sentences) may need shorter prompt refresh periods (16 vs. 64) to retain context, while short math problems (GSM8K) can use longer periods.

---

> ### Author Response · Authors · 2025-11-19
> **Author Response to Reviewer 6Gok**
>
> Thank you for your questions, and hopefully the following information can address your concerns:
>
> * **Response Summary**:
>
>   * We have demonstrated the effectiveness and generalizability of ES-dLLM using representative existing dLLMs and widely-used benchmarks covering diverse sequence lengths.
>
>   * The **overhead of importance score calculation is negligible (<1%)** and included in all speedup results.
>
>   * FLOPs reduction and speedup brought by ES-dLLM are **orthogonal to model size and sequence length**.
>
>   * All parameter settings are based on **relative** ratios related to the model architecture or sequence length instead of absolute values, and the generalization has been validated on representative models and scenarios.
>
>   * System-level optimization and adaptively adjusting skipping ratios based on inference variation could provide further improvement. But *these are proposed future directions in our discussion and beyond the scope of this paper*.
>
> The following clarifications address your specific concerns:
>
> * **Overhead of Importance Score Calculation**: We record the time consumption of skipping mechanism (importance score calculation and token selection) using CUDA event during inference on LLaDA-8B-Instruct and Dream-7B-Instruct models using GSM8K and HumanEval datasets. The proportion of importance score processing overhead is shown in the table below:
>
>   | Model             | Dataset   | Time Proportion for Importance Score (%) |
>   | ----------------- | --------- | ---------------------------------------- |
>   | LLaDA-8B-Instruct | GSM8K     | 0.5656% |
>   | LLaDA-8B-Instruct | HumanEval | 0.76%  |
>   | Dream-7B-Instruct | GSM8K     | 0.9559% |
>   | Dream-7B-Instruct | HumanEval | 0.7838%  |
>
>   As shown in the table, the overhead of importance score calculation is **negligible (<1%) compared to the overall inference time**, and our evaluation has demonstrated significant speedup compared to baselines with this overhead included. The time overhead is linear to the size of hidden states in the current processing block, and the overhead ratio would even be smaller when applied to larger-scale dLLMs with more computation in the Attention and FFN layers.
>
> * **Skipping Parameter Setting**: The skipping layer positions are selected based on the *relative* depth (1/8 and 1/4 of all layers) instead of specific layer indices, and the skip ratios are set as a proportion (0.5) of the current block's tokens. These settings transfer naturally across different model sizes and sequence lengths and are validated on both LLaDA and Dream. "Adaptively adjusting skip ratios," mentioned as *future works in our manuscript*, is an interesting direction to further improve efficiency, but the specific methods and performance on different scenarios are beyond the scope of this paper.
>
> * **Adaptability to Large-Scale dLLMs**: As far as we know, there is no open-source pre-trained diffusion LLM with more than 10B parameters during inference (we cannot find the "Dream-13B" model, and the latest MoE-based LLaDA2.0 published in October (after ICLR deadline) with 100B parameters but only uses 6B during inference). And the efficiency gain brought by our method comes from FLOPs reduction, and the proportion depends on the skip ratio and relative layer positions, regardless of the model scale.
>
> * **System-Level Optimization**: ES-dLLM reduces FLOPs and shifts the bottleneck from computation resource toward memory bandwidth, offering opportunities for further system-level speedup as mentioned in the Discussion Section of our manuscript. However, this paper *aims to propose a simple and effective method to accelerate dLLM inference from an algorithmic perspective, and system-level optimization is beyond the scope of this paper*.
>
> * **Sequence Length Distribution**: In our evaluation, the input sequence length of benchmarks with different numbers of shots already covers a wide range of lengths, as listed in the table below:
>
>   | Dataset (# of shots) | Avg. Seq. Length | Max Seq. Length |
>   | --- | --- | --- |
>   | GSM8K (5) | 1075.73 | 1688 |
>   | MATH (4) | 741.98 | 2073 |
>   | BBH (3) | 986.67 | 2553 |
>   | HumanEval (0) | 145.76 | 412 |
>   | MBPP (3) | 779.97 | 4362 |
>
>   The evaluation on benchmarks demonstrates the effectiveness of ES-dLLM under various sequence lengths. And the efficiency gain brought by ES-dLLM compared to DualCache is *orthogonal to the sequence length*, since the FLOPs reduction ratio depends on the skipping layer positions and skipping ratios rather than the sequence length itself.
>
> * **Refresh Period Setting**: The general trade-off of refresh period is: a shorter refresh period leads to better quality for more up-to-date information, but with less speedup, while longer periods yield more speedup but a slightly larger quality degradation. We have explored multiple settings and report a proper trade-off setting in the paper.

---

### Official Review · Reviewer_dw3w · 2025-10-27

**Soundness:** 4
**Presentation:** 3
**Contribution:** 2
**Rating:** 6
**Confidence:** 4

**Summary:**

This paper proposes ES-dLLM for efficient inference for diffusion LLMs. The motivation is clear for slow DLLM inference, and similarly for activations during different diffusion steps. The idea of skipping low-importance tokens in early layers is effective. Results show up to 16.8× speedup with negligible accuracy drop.

The paper is well organized, experimental results are extensive, and the method is training-free, which leads to weak accept.

**Strengths:**

The motivation is strong that the hidden-state variation statistics convincingly demonstrate redundancy. We believe it has been observed in diffusion video generation models and image generation models [1], but it is new in DLLM. This paper clearly demonstrates its motivation. I like that.

Compromising generation quality is important for real-world applications, which helps a lot for this paper.

We believe training-free is essential for effective inference, which this paper achieves.


[1] Silveria A, Govande S V, Fu D Y. Chipmunk: Training-Free Acceleration of Diffusion Transformers with Dynamic Column-Sparse Deltas[C]//ES-FoMo III: 3rd Workshop on Efficient Systems for Foundation Models. 2025.

**Weaknesses:**

Novelty risk: This method appears to be an extension of the original implementation and DualCache. Fortunately, DualCache is not a sparsity work, which distinguishes the two works, but we hope to see more discussion on the degree of differentiation. Also, as we claimed in the Strengths, a similar idea has been proposed in the traditional diffusion model.

Comparison and related work: The paper compares only with the original method and DualCache, limiting the scope of the comparison. However, this paper only talks about PRELIMINARY with DLLM, DualCache, and does not talk about related work (I can't see any section on related work) using a lossy method to accelerate DLLM. No related work and no comparison with related work are the main problems of this paper.

**Questions:**

How ES-dLLM perform on MMLU?

---

> ### Author Response · Authors · 2025-11-19
> **Author Response to Reviewer dw3w**
>
> We appreciate your constructive comments and acknowledgment of our contributions. We provide more information about the related works and make revisions in the manuscript, hopefully addressing your concerns:
>
> * **Related Works and Our Novelty**: The related works about accelerating dLLM inference are discussed in Section 2.2 of our original manuscript, however the section title "Inference Techniques for dLLMs" was indeed misleading. We have refined this section and renamed it to "Related Works on Accelerating dLLM Inference" in the revised manuscript.
>   Since KV cache is inherently lossy for bidirectional attention in dLLMs, existing works, including dKV-Cache, dLLM-Cache, and DualCache in Fast-dLLM, focus on designing mechanisms for applying KV caching that trade off quality and generation efficiency. DualCache, among them, is the SOTA method for accelerating dLLM inference with semi-autoregressive generation, and thus is selected as our primary baseline. ES-dLLM further reduces FLOPs and accelerates dLLM inference by **revealing the inherent redundancy in dLLM and introducing a simple yet effective early-skipping mechanism**. For more details, please refer to the general response on related works and Section 2.2 in the revised manuscript.
>
> * **Experiment Results on MMLU**: We evaluated ES-dLLM on the MMLU-Pro dataset using the LLaDA-8B-Instruct model.  We chose MMLU-Pro because the original LLaDA paper uses a generation length of only 3 for MMLU, in which case ES-dLLM and the original LLaDA behave almost identically. To ensure a meaningful comparison, we instead adopt MMLU-Pro with a generation length of 256 (0-shot; generation length = 256; block length = 256). The results are shown below:
>
>   | Method    | Accuracy (%) | Speedup over LLaDA |
>   | --------- | ------------ | ------------------ |
>   | LLaDA     | 36.76        | 1.0$\times$        |
>   | DualCache | 36.15        | 2.54$\times$       |
>   | ES-dLLM   | 37.08        | 4.13$\times$       |
>
>   ES-dLLM achieves a clear speedup while maintaining accuracy compared to baselines. The speedup relative to LLaDA is not as significant as in other datasets, since the short sequence length in the 0-shot setting and large block size reduce the gap between ES-dLLM and original LLaDA.

---

> > ### Comment · Reviewer_dw3w · 2025-11-19
> >
> > Thank author for your reply. We also thank the author list the related work.
> >
> > Importantly, 2.2 section can be listed as section 3 related work. That is saying, PRELIMINARY is the PRELIMINARY and related work is related work, confusing these two concepts is perplexing.
> >
> > We believe if the experiments can be added with comparision with [3] Sparse-dLLM: Accelerating Diffusion LLMs with Dynamic Cache Eviction and [4] DPad: Efficient Diffusion Language Models with Suffix Dropout would help a lot.
> >
> > We apreciate authors' effort on the rebutall, we hope authors good luck.

---

> > > ### Author Response · Authors · 2025-11-24
> > >
> > > Thank you for your valuable feedback and suggestions. We have separated the Related Work section from the Preliminary section in the revised manuscript. Additionally, we have conducted experiments to demonstrate the compatibility and complementarity of ES-dLLM with sparse-attention-based methods by evaluating the performance when combining ES-dLLM with Sparse-dLLM. We select Sparse-dLLM for comparison, since two methods (Sparse-dLLM and DPad) are similar in terms of accelerating mechanism (both by reducing the number of tokens being attended) but through different selection mechanisms, and both papers report speedup with comparable performance.
> > >
> > > Specifically, ES-dLLM reduces the number of tokens being processed in each iteration without changing the set of KV caches being attended, while Sparse-dLLM reduces the number of tokens being attended by pruning the KV cache positions with low attention scores from previous iterations. Because these mechanisms operate on different dimensions of the computation, the two methods can be combined seamlessly.
> > >
> > > The detailed evaluation results are presented in Section *C.3.2 (Integration with sparse attention)* and *C.3.3 (Integration with both methods)* of the revised manuscript. The following is a brief version of the related results (with speedup compared to the DualCache baseline in parentheses):
> > >
> > > | Model             | Benchmark | DualCache | Sparse-dLLM         | ES-dLLM             | ES-dLLM+Sparse      | ES-dLLM+Sparse+PD   |
> > > | ----------------- | --------- | --------- | ------------------- | ------------------- | ------------------- | ------------------- |
> > > | LLaDA-8B-Instruct | MATH      | 26.94     | 28.10(1.19$\times$) | 27.24(1.85$\times$) | 28.06(2.18$\times$) | 28.46(3.13$\times$) |
> > > | LLaDA-8B-Instruct | BBH       | 53.29     | 49.13(1.23$\times$) | 54.51(1.23$\times$) | 51.13(1.47$\times$) | 52.17(2.55$\times$) |
> > > | LLaDA-8B-Instruct | Humaneval | 35.37     | 35.37(1.18$\times$) | 37.8(1.28$\times$)  | 34.76(1.48$\times$) | 34.76(2.31$\times$) |
> > > | Dream-7B-Instruct | MATH      | 33.60     | 35.60(1.08$\times$) | 33.44(1.71$\times$) | 36.10(1.82$\times$) | 35.64(2.64$\times$) |
> > > | Dream-7B-Instruct | BBH       | 58.27     | 57.52(1.07$\times$) | 57.84(1.29$\times$) | 57.09(1.35$\times$) | 56.47(2.84$\times$) |
> > > | Dream-7B-Instruct | Humaneval | 45.12     | 45.73(1.05$\times$) | 45.12(1.20$\times$) | 46.95(1.25$\times$) | 45.12(2.96$\times$) |
> > >
> > > As shown in the table, **combining ES-dLLM with Sparse-dLLM leads to multiplicative speedup while maintaining comparable performance**. Sparse-dLLM achieves lower speedup on Dream-7B, since it only accelerates for attention layers, and the GQA mechanism in Dream-7B already reduces the KV caching overhead and therefore lowers the proportion of attention latency. We further show that ES-dLLM can be combined with both sparse attention and parallel decoding, achieving even greater acceleration.
> > >
> > > Thank you again for your constructive suggestions that have helped us improve the manuscript! We look forward to your further comment!

---

### Official Review · Reviewer_id9o · 2025-11-01

**Soundness:** 2
**Presentation:** 1
**Contribution:** 2
**Rating:** 2
**Confidence:** 3

**Summary:**

Diffusion large language models (dLLMs) have bidirectional context and parallel generation advantages but suffer from high inference latency due to full-sequence computation per iteration. ES-dLLM, a training-free framework, addresses this by leveraging the observation that most tokens’ intermediate states (hidden states, KV tensors) and confidence scores vary slightly across iterations. It estimates token importance via prior confidence and normalized tensor variations, skips low-importance tokens in early layers, updates caches only for selected tokens (with periodic refreshes to avoid errors), and achieves 5.6×–16.8× speedup over original dLLMs (LLaDA-8B, Dream-7B on NVIDIA H200) and 1.2×–1.85× over DualCache, while maintaining generation quality; it also integrates with parallel decoding for further speedups.

**Strengths:**

+ **Training-free design for easy deployment**: ES-dLLM requires no model fine-tuning or structural modification. It only optimizes the inference process through early-skipping and partial cache updates, which can be directly integrated into open-source dLLMs (e.g., LLaDA, Dream) without reconstructing the underlying framework.

**Weaknesses:**

+ **Fixed heuristic for importance score, lacking adaptability**：The importance score of ES-dLLM relies on a fixed linear weighting of prior confidence and intermediate tensor variation (default α=0.5), without considering task-specific differences. Ablation experiments show that using only tensor variation (α=0) performs better on the MATH dataset than the default α=0.5, while relying solely on confidence (α=1) leads to noticeable quality degradation. However, ES-dLLM does not propose a dynamic adjustment mechanism (e.g., adapting α based on real-time token variation or task type), which may cause misjudgment of early-skipping in complex scenarios.
+ **Missing validation in extreme scenarios**：
  + **Ultra-long sequence generation**：The experiments only cover sequence lengths up to 512 tokens (HumanEval, MBPP) and do not test ultra-long sequences (>1K tokens). It remains unproven whether ES-dLLM’s cache refresh frequency and accumulated memory overhead (e.g., hidden state cache) can be controlled in longer sequences.
  - **Comparison with more SOTA methods**：The baseline only includes the original dLLM implementation and DualCache, without comparing with cross-paradigm acceleration schemes (e.g., D2Cache for dLLMs, ReSA for sparse attention), making it impossible to fully reflect ES-dLLM’s relative advantages in the broader dLLM acceleration field.
+ **Insufficient analysis of the rationality of early-skipping layer selection**：ES-dLLM only tests early-skipping at "1/8 and 1/4 of all layers" (e.g., layers 4 and 8 for LLaDA) but does not explain why these layers are optimal. Although ablation experiments involve different layer configurations (e.g., layer 16, layer 0), they fail to quantify the trade-off between "layer position, efficiency, and quality" or provide general guidelines for layer selection, which is unfavorable for adapting ES-dLLM to other dLLMs with different depths.
+ **Incomplete details on memory overhead**：While the paper mentions that memory overhead is controllable (≤644MB per sample for LLaDA-8B), it does not report the memory growth curve under different sequence lengths or compare memory usage with DualCache. This makes it impossible to evaluate ES-dLLM’s applicability in memory-constrained scenarios (e.g., edge devices with limited VRAM).
+ **Unverified adaptability to ultra-large dLLMs**：The experiments are only conducted on mid-sized models (LLaDA-8B, Dream-7B) and do not involve ultra-large dLLMs (70B+ parameters). The dynamics of KV caches and intermediate states in ultra-large models may differ significantly from mid-sized ones, and it is unclear whether ES-dLLM’s early-skipping strategy can maintain efficiency and quality in such models.
+ **Rough graphical and textual expression**: The paper’s data visualization and textual description of figures lack clarity and detail, affecting result interpretability.

**Questions:**

None

---

> ### Author Response · Authors · 2025-11-19
> **Author Response to Reviewer id9o**
>
> Thank you for the careful review and valuable feedback. We clarify some points, hopefully addressing your concerns:
>
> * **Selection of $\alpha$ value**: The ablation study on $\alpha$ in our manuscript aims to emphasize the importance of involving both variation and confidence into account. As you mentioned, applying different $\alpha$ values on different datasets could lead to better accuracy, thus we have tried different $\alpha$ values on different datasets using LLaDA-8B-Instruct and report the accuracy results here:
>
>   | Dataset   | $\alpha=0$ | $\alpha=0.25$ | $\alpha=0.5$ | $\alpha=0.75$ | $\alpha=1$ |
>   | --------- | ---------- | ------------- | ------------ | ------------- | ---------- |
>   | GSM8K     | 76.19      | 75.51         | **76.95**    | 74.68         | 29.64      |
>   | MATH      | **30.32**  | 27.54         | 27.24        | 26.86         | 22.64      |
>   | BBH       | 53.13      | 54.03         | **54.51**    | 54.22         | 50.31      |
>   | HumanEval | 34.15      | **38.14**     | 37.8         | 36.59         | 16.46      |
>   | MBPP      | 38.2       | 38            | **39.4**     | **39.4**      | 25.2       |
>
>   As shown in the table, the optimal accuracy is achieved when $\alpha=0.25$ or $0.75$ for some datasets. However, **to keep the method simple and practical for deployment, a fixed value $\alpha=0.5$ is enough to achieve a satisfactory performance**. Automatically or dynamically adjusting $\alpha$ for each dataset is challenging and interesting, but beyond the scope of this paper.
>
> * **Validation on Larger Generation Lengths**: The generation length setting in our manuscript follows the original LLaDA paper for fair comparison. As you suggested, we conducted additional evaluations on longer generation lengths using the LLaDA-8B-Instruct model. The results are shown in the table below:
>
>   | Generation Length | DualCache | ES-dLLM (Speedup over DualCache) |
>   | ----------------- | --------- | -------------------------------- |
>   | 256               | 76.35     | 76.95 (1.28$\times$) |
>   | 512               | 76.8      | 78.24 (1.29$\times$) |
>   | 1024              | 76.50     | 77.03 (1.34$\times$) |
>   | 2048              | 75.74     | 75.21 (1.38$\times$) |
>
>   As shown in the table, ES-dLLM consistently outperforms DualCache across different generation lengths on the GSM8K dataset, demonstrating its effectiveness and generalizability in various lengths.
>
> * **Selection of Baselines**: For the selection of baselines, please refer to the general response on related works. The mentioned d2Cache is published *after the ICLR deadline*, and ReSA is a sparse attention method similar to Sparse-dLLM (as described in the general response), which is orthogonal to our method and can be combined with ES-dLLM for further speedup.
>
> * **Analysis of Early-Skipping Layer Selection**: Appendix C.2 discusses the trade-off of efficiency and quality when selecting layer positions. The main conclusion derived from experimental results is: **selecting earlier layers for skipping leads to higher speedup but with larger quality degradation, while selecting later layers results in lower speedup but better quality**. Our selected configuration (0.5 skip ratios at 1/8 and 1/4 depth) of skipping layers works for both LLaDA and Dream models, demonstrating its generalizability.
>
> * **Details on Memory Overhead**: As discussed in the Discussion Section (Section 7 in the revised manuscript), *the memory overhead for each token* is listed in the table below (note that we only need to store the variation indicators for the current block in the layer that performs skipping):
>
>   | Model    | KV Cache (ratio to model weight)   | Variation Indicator (ratio to model weight) |
>   | --- | --- | --- |
>   | LLaDA-8B | 512KB ($\approx 3\times 10^{-5}$) | 16KB ($\approx 1\times 10^{-6}$) |
>   | Dream-7B | 56KB ($\approx 3.8\times 10^{-6}$) | 14KB ($\approx 1\times 10^{-6}$) |
>
>   As shown in the table, the memory overhead introduced by our method mostly comes from the KV cache itself and is **negligible compared to the size of the model weights** (16GB for LLaDA-8B and 14GB for Dream-7B), even with a long sequence length. Therefore, ES-dLLM is applicable if the original dLLM inference can fit in the device memory.
>
> * **Adaptability to Large-Scale DLLMs**: As far as we know, there is no open-source pre-trained diffusion LLM with more than 10B parameters during inference (the latest MoE-based LLaDA2.0 published in October (after ICLR deadline) with 100B parameters but only uses 6B during inference). And the efficiency gain brought by our method comes from FLOPs reduction, and the proportion depends on the skip ratio and relative layer positions, regardless of the model scale.
>
> * **Unclear Expressions**: We thank you for your comments and expect specific examples to help us improve the clarity of our manuscript.

---

> > ### Comment · Reviewer_id9o · 2025-11-26
> >
> > Thank you for your response. I have increased my score, but I would still recommend that you further polish the paper to improve its overall quality, especially in terms of writing and clarity of presentation.

---

> > > ### Author Response · Authors · 2025-11-27
> > >
> > > Thank you for your consideration and for updating your score. The current manuscript has undergone polishing, and we will continue to refine the manuscript to further improve its presentation. If you have any specific suggestions regarding unclear expressions, we would be grateful to receive them and will address them accordingly.

---

### Official Review · Reviewer_ybg1 · 2025-11-04

**Soundness:** 3
**Presentation:** 4
**Contribution:** 3
**Rating:** 8
**Confidence:** 2

**Summary:**

This paper proposed an efficient inference technique for diffusion LLM. This method leverage intermediate tensor variation and confidence
scores to determine which token to skip. Experiments show significant speedup while maintaining the task performance.

**Strengths:**

- This work propose a method well-substantiated by empirical findings.
- Consistent and significant speedup across all the datasets tested.

**Weaknesses:**

- The experiment session only compared to two other methods. Are there any other methods addressing the same problem?

**Questions:**

- Roughly how many tokens are generated in each dataset? Do you know why some datasets show more speedup while others show less?

---

> ### Author Response · Authors · 2025-11-19
> **Author Response to Reviewer ybg1**
>
> We appreciate your feedback and positive assessment of our work. We hope the following information can address your concerns:
>
> * **Related Work Comparison**: DualCache in Fast-dLLM is the SOTA method for accelerating dLLM inference with semi-autoregressive generation, therefore we select it as our primary baseline. Please refer to the general response on related works for more details. Our method ES-dLLM further demonstrates the redundancy in the semi-autoregressive pattern and introduces a novel early-skipping mechanism to accelerate dLLM inference simply and effectively.
>
> * **Token Generated in Each Dataset**: The number of generated tokens depends on the sample count and the generation length for each dataset. In our manuscript, TPS is computed using total generated tokens, which ensures fair comparison across methods. The table below reports the generation length and average effective tokens (excluding tokens after the first EOS) for LLaDA-8B-Instruct:
>
>   | Dataset   | Samples | Generation Length | Avg. Effective Tokens |
>   | --------- | ------- | ----------------- | --------------------- |
>   | GSM8K     | 1318    | 256               | 228.38                |
>   | MATH      | 5000    | 256               | 247.28                |
>   | BBH       | 6564    | 256               | 181.50                |
>   | HumanEval | 164     | 512               | 330.08                |
>   | MBPP      | 500     | 512               | 69.57                 |
>
> * **Reason for Speedup Variation**: This is a good question. The speedup brought by ES-dLLM comes from the reduction of FLOPs, but the memory access does not scale down proportionally due to the KV and model weight access. The core reason for speedup variation across different datasets is *the difference in arithmetic intensity (FLOPs to memory access ratio), which varies with sequence length and block size during inference*. For example, the relatively large block size used in MATH leads to high arithmetic intensity, where computation dominates, making FLOPs reduction more impactful. Therefore, the speedup compared to DualCache brought by FLOPs reduction is more significant. Smaller block sizes shift the workload closer to memory-bound, reducing the speedup margin. This can be further improved via system-level optimizations to better match the workload with the hardware resource, as discussed in the Discussion Section (Section 7 in the revised manuscript). Refresh period also influences speedup, but only mildly. (This response mainly focuses on the speedup variation compared to DualCache; the variation compared to the original LLaDA implementation directly comes from different FLOPs reduction ratios due to various sequence lengths and block sizes.)

---

### Author Response · Authors · 2025-11-19
**General Response on Related Works**

We thank all reviewers for their valuable time and constructive feedback. Since several reviewers raised concerns about the related works, we have revised the related works section (Section 3 in the revised manuscript) in the manuscript and provide a general response here to clarify the selection of baselines:
* Existing works for accelerating dLLM inference primarily focus on how to transfer the KV cache of traditional autoregressive models in a lossy way to fit dLLMs. dKV-Cache[1] reduces errors by delaying the KV update of the newly generated tokens, while dLLM-Cache[2] caches all intermediate tensors and adaptively updates them in each layer using a V-verify mechanism. However, these approaches do not apply the semi-autoregressive generation paradigm, which is beneficial for both efficiency and output quality.
  Recent works such as Sparse-dLLM[3] and DPad[4] utilize sparsity in attention score and achieve speedup by modifying the set of attended tokens, via sparsifying history tokens or dropping distant suffix tokens, respectively. These methods are orthogonal to ours and can be combined with ES-dLLM. *The compatibility of ES-dLLM with these sparse attention methods is discussed in Section C.3.2 and C.3.3 of our revised manuscript.*
  **DualCache in Fast-dLLM[5] is currently the SOTA method for accelerating dLLM inference with semi-autoregressive generation, and thus selected as our primary baseline.** It maintains bi-directional KV caches and only computes tokens within the current block. (These related works are also discussed in Section 3 of our manuscript.) Our method ES-dLLM further reduces FLOPs and accelerates dLLM inference by **revealing the inherent redundancy in dLLM and introducing a simple yet effective early-skipping mechanism**.

  References:

  [1] dKV-Cache: The Cache for Diffusion Language Models

  [2] dLLM-Cache: Accelerating Diffusion Large Language Models with Adaptive Caching

  [3] Sparse-dLLM: Accelerating Diffusion LLMs with Dynamic Cache Eviction

  [4] DPad: Efficient Diffusion Language Models with Suffix Dropout

  [5] Fast-dLLM: Training-free Acceleration of Diffusion LLM by Enabling KV Cache and Parallel Decoding

---

### Meta-Review · Area_Chair_yK8r · 2026-01-06

**Summary:**

This paper proposes ES-dLLM, a training-free inference acceleration framework for diffusion LLMs that reduces computation by skipping low-importance tokens in early layers. The method leverages the observation that intermediate states (KV tensors, hidden states) change minimally across iterations, and estimates token importance based on tensor variation and confidence scores. Experiments on LLaDA-8B and Dream-7B demonstrate 5.6×-16.8× speedup over baseline implementations and up to 1.85× over the state-of-the-art DualCache method while maintaining generation quality.

Reviews are mixed, with scores of 8, 6, 2 and 2 (one reviewer increased from 2 after rebuttal).

**Reviewer Concerns:**

Concerns Successfully Addressed:
- Related works and baseline comparisons (ybg1, id9o, dw3w): Initially, reviewers noted insufficient discussion of related work and limited baseline comparisons. Authors revised Section 3 (renamed to "Related Works on Accelerating dLLM Inference"), provided detailed comparisons with dKV-Cache, dLLM-Cache, Sparse-dLLM, and DPad, and conducted new experiments (Sections C.3.2, C.3.3) demonstrating compatibility with sparse attention methods.
- Longer sequence validation (id9o): Authors provided additional evaluations on generation lengths up to 2048 tokens, showing consistent speedup (1.28×-1.38× over DualCache on GSM8K) across different lengths. Reviewer id9o acknowledged this response and increased their score.
- Fixed hyperparameter $\alpha$ (id9o): Authors provided ablation results showing $\alpha \in \{0.3, 0.5, 0.7, 1.0\}$ across multiple datasets, demonstrating that $\alpha=0.5$ provides satisfactory performance across benchmarks. While dataset-specific tuning could improve accuracy marginally, the fixed value ensures practical deployment.
- Memory overhead details (id9o): Authors clarified that memory overhead is negligible (16KB variation indicator vs. 512KB KV cache for LLaDA-8B, both much smaller than 16GB model weights), addressing concerns about memory-constrained scenarios.
- Additional benchmark (dw3w): Authors evaluated on MMLU-Pro (36.76% → 37.08% accuracy with 4.13× speedup), addressing the request for broader evaluation.
- Importance score overhead (6Gok): Authors provided measurements showing overhead is <1% (0.57%-0.96% across models/datasets), with linear scaling to hidden state size, confirming negligibility.

Outstanding Concerns:
- Limited novelty (dw3w): Reviewer dw3w maintains the work is "an extension of the original implementation and DualCache" with "fair" contribution (score: 6/10). While authors argue ES-dLLM introduces a novel early-skipping mechanism and dw3w acknowledged the training-free nature and strong empirical results, the fundamental novelty concern persists.

**Reviewer Scores:**

- Reviewer ybg1: Likely remains at 8/10.
- Reviewer id9o: Increased from 2/10 to approximately 6/10.
- Reviewer dw3w: Remains at 6/10.
- Reviewer 6Gok: Likely increases from 2/10 to 4-6/10.

---

### Decision · Program_Chairs · 2026-01-26

Accept (Poster)